# Comparison of Minimum Inhibitory Concentrations of Selected Antimicrobials for Non-Aureus *Staphylococci*, *Enterococci*, *Lactococci*, and *Streptococci* Isolated from Milk Samples of Cows with Clinical Mastitis

**DOI:** 10.3390/antibiotics13010091

**Published:** 2024-01-18

**Authors:** Quinn K. Kolar, Juliano L. Goncalves, Ronald J. Erskine, Pamela L. Ruegg

**Affiliations:** 1Department of Animal Science, Michigan State University, East Lansing, MI 48824, USA; kolarqui@msu.edu; 2Department of Large Animal Clinical Sciences, Michigan State University, East Lansing, MI 48824, USA; goncal25@msu.edu (J.L.G.); erskine@msu.edu (R.J.E.)

**Keywords:** antibiotics, mastitis, Gram-positive, dairy

## Abstract

The objective of this study was to compare the minimum inhibitory concentrations of antimicrobials included in a commercial broth microdilution panel among Gram-positive pathogens that caused non-severe clinical mastitis on three Michigan dairy farms. Duplicate quarter milk samples were collected from eligible quarters of cows enrolled in a randomized clinical trial, cultured in a university laboratory, and identified using MALDI-TOF. Etiologies were grouped by genus as *Enterococcus* species (n = 11), *Lactococcus* species (n = 44), non-aureus *Staphylococcus* species (n = 39), or *Streptococcus* species (n = 25). Minimum inhibitory concentrations (MICs) were determined using the mastitis panel of a commercially available broth microdilution test. In vitro susceptibility was determined using approved guidelines and included breakpoints for mastitis pathogens, or when not available, breakpoints from other species. Most isolates were inhibited at or below breakpoints that demonstrated in vitro susceptibility. The proportions of susceptible isolates varied among pathogens for pirlimycin, penicillin, and tetracycline. The greatest proportion of resistance was observed for pirlimycin, tetracycline, and sulfadimethoxine. Survival analysis was performed to evaluate differences in MICs among pathogen groups. MIC values varied among pathogens for ceftiofur, cephalothin, erythromycin, penicillin, pirlimycin, and tetracycline. However, nearly all isolates were susceptible to ceftiofur and cephalothin, indicating that pathogen differences in MIC are not likely clinically relevant, as these are the two most commonly administered mastitis treatments in the United States. While differences in vitro susceptibility were observed for some antimicrobials, susceptibility was high to cephalosporin-based IMM treatments that are most commonly used and did not vary among pathogens.

## 1. Introduction

Prevention and treatment of mastitis are the primary reasons for the administration of antimicrobials to adult dairy cows [1,2]. The use of antimicrobials in food-producing animals is increasingly scrutinized because of growing fears of antimicrobial resistance with an emphasis on reducing usage of critically important antimicrobials (CIAs) such as third-generation cephalosporins which belong to classes that are important for the treatment of human diseases [3]. Phenotypic susceptibility testing is one tool used to measure antimicrobial resistance (AMR) of bacterial pathogens. Susceptibility testing can be used to help make clinical decisions about the selection of antimicrobials and can be used to monitor the development of resistance over time. Although there is sparse evidence that AMR is driven by the intramammary (IMM) administration of antimicrobials [4,5], there is some evidence that systemic antibiotic treatments may contribute to AMR in non-aureus *Staphylococcci* (NAS) [6]. One mechanism to reduce the usage of CIA is to preferentially use narrow-spectrum antibiotics when possible [7]. While the results of phenotypic susceptibility testing are not strongly predictive of clinical outcomes [8,9], it is important that veterinarians can recommend antibiotic treatment using an appropriate spectrum of activity for pathogens that cause clinical mastitis (CM). However, the diversity of pathogens that cause clinical mastitis has changed considerably over the decades [10,11]. Most IMM antimicrobials available in the U.S. were developed to treat mastitis caused by *Staphylococci aureus* and some *Streptococci* spp., and have not been evaluated for efficacy against NAS, *Enterococcus* spp., or *Lactococcus* spp., which are increasingly isolated from mastitis [11]. The objective of this study was to identify NAS and other cocci to compare the MICs of selected Gram-positive pathogens that caused non-severe CM on Michigan dairy farms. We hypothesized that *Streptococci*, *Enterococci*, *Lactococci*, and non-aureus *staphylococci* (NAS) would have different MICs and susceptibilities to the tested antimicrobials.

## 2. Results

Of 148 cases of CM enrolled from MI herds, 20 cases were excluded because they were in a pathogen group that had fewer than 10 isolates (Staph aureus (n = 4 cases), *Trueperella pyogenes* (n = 2 cases), *Bacillus* species (n = 3 cases), *Corynebacterium* spp. (n = 1 case), Pseudoarthrobacter spp. (n = 1 case), Paenibacillus species (n = 1 case), *Pseudomonas* spp. (n = 3 cases)), or because two pathogens (n = 3), or *Yeast* spp. (n = 2) were recovered from the milk sample. Of the 128 remaining cases, bacteria from 1 case did not survive cryopreservation, and bacteria from an additional 19 cases were not included because the initial duplicate milk sample was contaminated (n = 6), had non-significant growth (n = 11), or was missing (n = 2). Cases used for MIC testing (n = 108) included a diverse group of Gram-positive cocci. *Lactococcus* spp. were most prevalent, with half of the cases caused by *Lactococcus* spp. or *Enterococcus* spp. (Table 1). 

Susceptibility to antimicrobials included in the commercial panel ranged from 6.8% (susceptibility of *Lactococci* to penicillin) to 100% (Table 2). All isolates demonstrated in vitro susceptibility to oxacillin, and the majority were susceptible to erythromycin (Table 2). The susceptibility of NAS to commercially available IMM products ranged from 79% to >90%, but were less for sulfadimethoxine and for tetracycline (Table 2). The susceptibility of Strep. to IMM antibiotics available in the United States were moderate (penicillin), to high (ampicillin) (ceftiofur, cephapirin, and the combination of penicillin and novobiocin) (Table 2). However, the minority of *Streptococci* species were susceptible to sulfadimethoxine or tetracycline (Table 2). Only a small proportion of *Lactococci* demonstrated in vitro susceptibility to penicillin, while the majority were susceptible to ampicillin and all were susceptible to ceftiofur, cephapirin, and the combination of penicillin and novobiocin (Table 2). All *Enterococci* species were susceptible to all commercially available IMM antibiotics, but few were susceptible to tetracycline or sulfadimethoxine (Table 2). 

Susceptibility to ampicillin was not associated with the pathogen group (*p* = 0.165), and no potentially confounding variables met inclusion criteria for that model. Susceptibility to penicillin varied among pathogen groups (*p* < 0.001) and was influenced by parity (*p* = 0.04). As compared with NAS, the odds of susceptibility to penicillin were less for *Lactococci* (OR = 0.014, 95% CI [0.002, 0.094]) and *Streptococci* (OR = 0.067, 95% CI [0.011, 0.39]). For penicillin, isolates obtained from cases that occurred in primiparous animals were less likely to be susceptible isolates obtained from multiparous animals (OR = 0.25, 95% CI [0.06, 0.96]).

Susceptibility to tetracycline varied among pathogen groups (*p* = 0.02) and was influenced by a prior occurrence of subclinical mastitis (*p* = 0.04). When SCC > 150,000 cells/mL preceding the CM case, the odds of susceptibility to tetracycline decreased (OR = 0.30, 95% CI [0.10, 0.92]). Susceptibility to sulfadimethoxine was not associated with pathogen group (*p* = 0.12), and no potential confounder met the inclusion criteria for the model. Susceptibility to pirlimycin varied among pathogen groups (*p* = 0.01). As compared with NAS, *Lactococci* species were less likely to be susceptible to pirlimycin (OR = 0.11, 95% CI [0.01, 0.89]). Overall, the *Lactococci* group had the lowest proportion of susceptibility to pirlimycin, with 40.9% not inhibited at the greatest tested concentration (Table 2).

For cephalothin, oxacillin, penicillin novobiocin, and pirlimycin, the MIC_50_ was the same for all four pathogen groups (Table 2). For penicillin, each pathogen group had a different MIC_50_. For ampicillin, the MIC_50_ was the least tested concentration (0.12 μg/mL) for all pathogen groups except Lact (0.25 μg/mL, Table 2). For ceftiofur, the MIC_50_ was the least tested concentration (0.5 μg/mL) for all pathogen groups except for NAS (1 μg/mL, Table 2). The overall MIC_90_ could not be calculated for tetracycline and sulfadimethoxine (for all pathogens), pirlimycin (Strep. and Lact.), and erythromycin (Strep.) groups because greater than 10% of these isolates were not inhibited at the greatest concentration tested in this commercially available panel (Table 2). 

Associations between pathogen group and MICs were evaluated for all 10 antimicrobials using survival curves. While all isolates were inhibited below the breakpoint for cephalothin, differences in MIC were observed among pathogens (log-rank and Wilcoxon, *p* < 0.001). 

Based on survival analysis, we observed variations among MIC values for ceftiofur for different pathogen groups, (log-rank and Wilcoxon, *p* ≤ 0.001). Differences in survival curves for MIC (log-rank and Wilcoxon, *p* ≤ 0.009) were observed for tetracycline, penicillin, and pirlimycin, and significant differences were observed in the susceptibility. Of all the antimicrobials tested, only oxacillin, penicillin novobiocin, and sulfadimethoxine had no variation in MIC when assessed using log-rank (*p* > 0.119) and Wilcoxon (*p* > 0.13, Table 2).

No differences in the equality of strata at greater antimicrobial concentrations were observed for ampicillin (Figure 1A; log-rank *p* = 0.32), but inhibition of growth varied among pathogen groups at lower concentrations (Wilcoxon, *p* = 0.03). Inhibition of bacterial growth varied among pathogen groups for pirlimycin (Figure 1B). The Lacto. group had the highest proportion of isolates not inhibited at the greatest tested concentration (Figure 1B). 

There were no differences among pathogen groups in the proportion of isolates inhibited by sulfadimethoxine (Figure 2A); however, for many pathogens, we were unable to determine the MIC_50_ or MIC_90_ because many isolates were not inhibited at the greatest tested concentration (Table 2). Similar to sulfadimethoxine, (Figure 2B), a large proportion of isolates were not inhibited at the greatest concentration of tetracycline (Table 2).

Cephalothin is the class representative for first-generation cephalosporins. Few isolates were resistant to either ceftiofur or cephalothin (Table 2), although the MIC differed among Gram-positive pathogens (Figure 3A). For example, 80% of NAS were inhibited at the least tested concentration of penicillin as compared with 7% of *Lactococci* (Figure 3B). 

## 3. Discussion

In recent years, a greater variety of pathogens has been recovered from milk samples collected from cows diagnosed with clinical mastitis; however, environmental *Streptococci* spp. are typically commonly diagnosed [15,16]. For decades, based on the use of classical microbiological techniques, Gram-positive catalase negative cocci recovered from bovine milk samples were diagnosed as *Streptococcus* spp. [17]. However, the use of the matrix-assisted laser desorption/ionization time of flight (MALDI-TOF [10] and other improved laboratory methods [18] has led to more precise diagnoses of mastitis pathogens which has coincided with greater numbers of *Lactococci* spp. identified from milk samples of cows affected with mastitis [19]. While *Lactococci* spp., present with non-specific clinical signs that are indistinguishable from other intramammary infections, treatment protocols for these infections are not well defined. No IMM antimicrobials that are currently marketed in the U.S. have labeled efficacy for *Lactococci* or *Enterococci*; thus, more data about potential differences in susceptibility of these pathogens are useful. Only IMM antimicrobials are approved for the treatment of mastitis in the U.S., and all are classified as β-lactams. Thus, there are limited antimicrobial choices for the consideration of bovine mastitis treatment. The cases that were enrolled in this study were uncomplicated, non-severe cases that were detected by farm workers on commercial dairy farms and are typical of the majority of mastitis cases occurring on commercial dairy farms in the U.S. [16]. The isolates that were used in this study were tested against 10 class representative antibiotics that are included in the most common commercially available MIC panel used in reference laboratories for mastitis. Of these antibiotics, four are currently marketed in the U.S. as IMM treatments for clinical mastitis (ampicillin, ceftiofur, cephalothin, penicillin); one is available as an intramammary dry cow product (penicillin novobiocin); two are approved for IMM treatment but no longer marketed in the U.S. (pirlimycin and erythromycin); one (tetracycline) is labeled for systemic administration in dairy cows but not labeled for the treatment of mastitis, although extra-label usage is allowed under veterinary supervision; and one (sulfadimethoxine) is only labeled for the treatment of pneumonia and footrot in dairy cows (no extra-label usage of this product is allowed). As few antibiotics are approved to treat mastitis in U.S. dairy herds, most cases are treated using either first- or third-generation cephalosporins [1,2]. 

There were several encouraging results from this study. Oxacillin is included in the commercial panel as an initial screen for possible resistance to methicillin, which then must be confirmed by the identification of *mecA* or *mecC* genes. *mecA* or *mecC* genes are frequently associated with antimicrobial resistance, and have been identified in the NAS group [20]. Notably, 100% of the isolates used in this study were susceptible to oxacillin (Table 2); this finding is consistent with studies that have demonstrated only a sparse development of methicillin resistance in U.S. dairy herds [21]. Similarly, all but one of the isolates were susceptible to the combination of penicillin and novobiocin (Table 2). Based on defined breakpoints, antimicrobial susceptibility results are classified as susceptible, intermediate, or resistant, with intermediate implying that clinical efficacy can be achieved if the antimicrobial reaches therapeutic concentration [13]. Based on CLSI guidelines [12,13], all isolates were susceptible to ceftiofur, except for two NAS isolates that were intermediate (Table 2), which is similar to previous studies [14,22]. Ceftiofur is the only third-generation cephalosporin licensed for IMM formulation which is a CIA and it is also the most commonly used IMM antibiotic [2]. Ceftiofur is available in the United States as both a dry cow and lactating cow formula. Our research shows that Gram-positive mastitis pathogens remain susceptible to ceftiofur in vitro, with identical rates of susceptibility that were seen from 1994 to 2000 in *Streptococci* [22]. 

As compared with other pathogen groups (Figure 1A), *Lactococci* were not inhibited until greater antimicrobial concentrations for ampicillin. Another study performed using *Lactococci lactis* subspecies *lactis* isolates from cows in New York and Minnesota used the European Food Safety Authority (EFSA) breakpoints to evaluate the antimicrobial susceptibility [23]. The EFSA has proposed breakpoints available for *Lactococcus lactis* ampicillin, penicillin, erythromycin, and tetracycline [23]. However, applying the CLSI guidelines [12,13] (<0.25 µg/mL) to the results found by Werner et al. (2014) [19], 95% of their isolates would have been susceptible to ampicillin compared with the 63.7% of isolates in our study. One possible reason for this could be species or even subspecies differences between the isolates. Werner et al. (2014) [19] were able to identify the isolates specifically as *Lactococcus lactis* subspecies *lactis* compared with our research, which contained 41% of isolates that were *Lactococcus garvieae*. 

In the United State, pirlimycin is no longer marketed. Previous research in Wisconsin (which grouped *Lactococci* spp. with other Strep-like-organisms) found that 71% of the isolates were susceptible to pirlimycin [14]. Additional work from Canada found similar rates (59.4%) of resistance to pirlimycin among *Lactococci* isolates from mastitis pathogens [24]. Werner et al. (2014) [19] used EFSA breakpoints, although EFSA does not have a breakpoint for pirlimycin. Applying the CLSI breakpoints to their results, 88% of the *Lactococci lactis* isolates would be classified as susceptible using the breakpoint of ≤2 µg/mL [19]. It has previously been proposed that *Lactococcus garvieae* are intrinsically resistant to clindamycin, and pirlimycin is a derivative of clindamycin [25]. In our study, 88% (n = 16) of the *Lactococcus garvieae* isolates were resistant to pirlimycin, while only 11.5% (n = 3) of the *Lactococcus lactis* isolates were resistant. Future research should evaluate AMR genes among the *Lactococci species* identified to look for possible interspecies transfer between *Lactococcus garvieae* and *Lactococcus lactis*.

The majority of the *Streptococcus*, *Lactococcus*, and *Enterococcus* isolates were not inhibited at the greatest tested concentration of sulfadimethoxine (Table 2). These results were interesting particularly because they directly contradicted the results found by Werner et al. [19], who reported that the majority of the isolates were susceptible at the least tested concentration (32 µg/mL). One possible reason for this discrepancy could be species differences, as 41% of the isolates used in our analysis were *Lactococcus garvieae* (Table 1) while all of the isolates reported by Werner et al., (2014) [19] were *Lactococcus lactis* subspecies *lactis*. Similar to sulfadimethoxine, a large percentage of isolates were not inhibited at the greatest tested concentration of tetracycline. Both antimicrobials are used in the United States to treat respiratory disease in dairy cattle and are not available as intramammary formulations, but are available for the systemic treatment of other bovine diseases. These two antimicrobials are also widely used in food animal medicine. Tetracyclines comprise the largest percentage (66%) of the total mass of medically important antimicrobials sold annually for use in food-producing animals in the USA [26]. Sulfonamides are also a large percentage (5%) of the total medically important antimicrobials sold for use in food-producing animals in the United States. Tetracyclines and sulfonamides are mostly administered in feed or water for other types of livestock (oral administration of these antibiotics is not allowed for lactating cows) and are sparingly used to treat dairy cows [1]. It is important to note that the extra-label usage of sulfonamides is not allowed in the U.S., and mastitis is not included on approved product labels. While the majority of CM cases in the United States are treated with IMM ceftiofur [2], IMM administration of this product accounts for <1% of the mass of all medically important antimicrobials used in 2020 [26]. Previous research concerning the resistance of bacteria causing CM has also documented high rates of resistance to sulfadimethoxine and tetracycline [14,24], Research on trends in antimicrobial susceptibility from 1994 to 2000 in Michigan found that 54.8% of *Strep uberis* and 39.8% of *Strep dysgalactiae* isolates were susceptible to tetracycline [22]. The CM isolates in this study collected two decades later confirm that tetracycline resistance remains common in *Streptococci* from CM isolates.

This research was limited by the available CLSI and EFSA guidelines. Currently, the CLSI guidelines have interpretative criteria for CM isolates for three antimicrobials (ceftiofur, penicillin/novobiocin, and pirlimycin) and only for limited pathogens, while the EFSA has breakpoints for ampicillin, penicillin, erythromycin, and tetracycline. For consistency, we used CLSI breakpoints and for the antimicrobial combinations and pathogens not listed; CLSI recommends using human MIC breakpoints. Although comparing the non-specific breakpoints for pathogen groups is the best available approach, it limits our understanding of the clinical role of different MIC breakpoints, especially for cephalothin and penicillin (Figure 3), where we are limited to a breakpoint from humans. The IMM formulation of penicillin is not commonly used to treat mastitis in the United States, with 0.8% of operations using it during the last survey [2]. Previous research in Michigan conducted from 1994 to 2000 found that 94.5% of *Streptococcus dysgalactiae* and *Streptococcus uberis* isolates were susceptible to penicillin [22]. In our study, only 44% of *Streptococci* and 6% of *Lactococci* were susceptible to penicillin, which is similar to recent research in Canada which reported that 22% of *Strep. uberis* and 0% of *Lactococcus* spp. were susceptible to penicillin [24]. In 2010, work from Switzerland demonstrated an in vitro a shift towards penicillin resistance in *Strep uberis* isolates [27]. Haenni et al. highlighted [27] that *Strep uberis* has a reservoir of mutated penicillin-binding protein mutations and that these genes could be possibly transmitted to other species. Future research to look for possible AMR gene transmission would be interesting.

## 4. Materials and Methods

Cows with non-severe CM were enrolled in a randomized clinical trial conducted on dairy farms located in Michigan (n = 3) and Minnesota (n = 1) during June 2019–March 2020. Cases of clinical mastitis were identified by trained farm workers who detected mastitis based on observations of abnormal foremilk and aseptically collected duplicate quarter milk samples following NMC procedures [28]. One milk sample was used to perform on-farm culture using a tri-plate containing selective agars; the case was enrolled in the clinical trial if significant growth was observed on Gram-positive agars of the selective media. The second milk sample was frozen and submitted to a university laboratory where identification was performed using matrix-assisted laser desorption/ionization time-of-flight (MALDI-TOF). Bacteria from the Michigan herds were cryopreserved at −80 °C, and only non-aureus *Staphylococci* species (NAS, n = 28), *Streptococci* species (Strep; n = 25), *Lactococci* species (Lact; n = 44), and *Enterococci* species (Enter; n = 11; Table 1) from cases enrolled in Michigan were used. Minimum inhibitory concentrations (MICs) were determined using the mastitis panel from a commercial broth microdilution test (Sensititre Vet Mastitis CMV1AMAF, Thermo Fisher Scientific) following guidelines of the Clinical and Laboratory Standards Institute (CLSI) [13]. Pure subcultures were grown from the frozen isolates, and broth microdilution was performed according to manufacturer’s instructions. Quality control was performed in accordance with CLSI using *S. aureus* (ATCC 29213), *Staphylococcus epidermidis* (ATCC 51625), *Lactococcus cremoris* (ATCC 19257), and *Lactococcus garvieae* (ATCC 43921). The control strain results were compliant with the quality control ranges.

Antimicrobial resistance was defined as the ability of the bacterial isolate to demonstrate growth at a defined concentration of the antimicrobial above a defined breakpoint [13]. Minimum inhibitory concentrations were defined as the least concentration of an antimicrobial that inhibited visible growth, while MIC_50_ and MIC_90_ were defined as the antimicrobial concentrations where 50% and 90% of isolates were inhibited, respectively. Breakpoints for susceptibility were defined using CLSI guidelines [12]. Bovine-mastitis-specific interpretative criteria were available for ceftiofur, penicillin and novobiocin, and pirlimycin [12]. The interpretative criteria for ampicillin, cephalothin, erythromycin, oxacillin, penicillin, and tetracycline were based on human criteria [12], which is consistent with the previous literature [14,24]. Current CLSI guidelines [12] do not have recommended breakpoints for sulfadimethoxine for any bovine mastitis pathogens; thus, consistent with previous studies, the susceptibility breakpoint was defined as ≤128 μg/mL [14]. As *Lactococci* spp. do not have their own category in the CLSI guidelines, the breakpoints for *Streptococci* spp. were used (Table 2). PROC GLIMMIX was used to test the hypothesis that susceptibility (binary outcome) varied among pathogen groups; means were adjusted using Tukey. *Enterococci* were not included in models for ampicillin and penicillin, as all isolates were inhibited at levels interpreted as susceptible (Table 2). However, CLSI guidelines [13] indicate that cephalosporins are not normally clinically effective for the treatment of *Enterococci* infections in animals. Similarly, models testing susceptibility for ceftiofur, cephalothin, erythromycin, oxacillin, and the combination of penicillin and novobiocin were not performed because >80% of all isolates were susceptible. For all models, univariate relationships between selected potential confounding variables and susceptibility were assessed, and variables that met inclusion criteria (*p* < 0.20) were offered to multivariable models. Variables considered for the models included season (warm or cool), days in milk (DIM) category (0–100 DIM, 101–200 DIM, or >201 DIM), parity (primiparous or multiparous), clinical mastitis history (defined as the occurrence of CM in any quarter of the cow during the 55 d before the case was enrolled), and subclinical mastitis history (defined as previous monthly DHIA SCC before the case > 150,000 cells/mL).

Survival analysis was performed to evaluate differences in MICs among pathogen groups using PROC LIFETEST in SAS (v. 9.4). Antimicrobial concentrations were used as “time”, while the event was defined as the inhibition of bacterial growth (Figure 3). Isolates that grew at the greatest concentration of antimicrobials included in the commercial panel were censored. Kaplan–Meier survival curves were produced for each of the antimicrobials and pathogen groups. The log-rank and Wilcoxon tests were used to evaluate the differences in survival among pathogen groups, with the null hypothesis that there were no differences in the survival curves of the strata (pathogen groups).

## 5. Conclusions

Overall, our results indicate that there are differences in the MICs of some antimicrobials among selected groups of Gram-positive mastitis pathogens, but also indicate that resistance to approved intramammary products is uncommon. It is well documented that MICs are not strongly predictive of clinical outcomes, which is likely a result of the complex interaction of bacterial pathogenesis, bovine immune response, and pharmacology of mastitis treatments [8,29]. However, overall knowledge about the susceptibility of mastitis pathogens to antibiotics can be used to inform initial treatment decisions and for surveillance about the emergence of resistance [30]. Many larger dairy farms are making treatment decisions for non-severe cases of CM by culturing milk using on-farm cultures, which usually differentiates results into categories such as “Gram-positive”, “Gram-negative”, or “culture negative”. Antibiotics are generally recommended for the treatment of Gram-positive infections. When making treatment decisions for CM, among available IMM products in the USA, the greatest rates of susceptibility were for products containing first- and third-generation cephalosporins; we found no differences in the susceptibility of pathogens for those antimicrobials, which suggests that the use of CIA is not essential for the effective treatment of Gram-positive intramammary infections. Future research should explore possible antimicrobial resistance genes in *Lactococci* spp., which have different MIC and susceptibility for many of the antimicrobials tested. Ultimately, while differences in in vitro susceptibility were observed for some antimicrobials, overall susceptibility to the most commonly used cephalosporin-based IMM treatments was high and did not vary among pathogens.

## Figures and Tables

**Figure 1 antibiotics-13-00091-f001:**
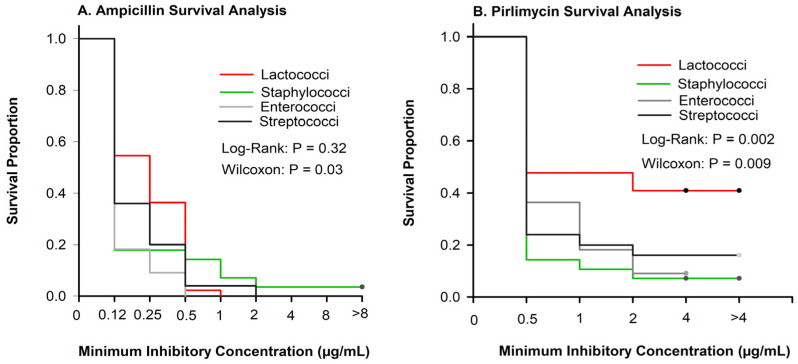
Kaplan–Meier plots showing survival proportions of *Lactococci* (n = 44), *Enterococci* (n = 11), *Staphylococci* (n = 28), and *Streptococci* (n = 25). (**A**) Inhibition of ampicillin varied at smaller concentrations (Wilcoxon (*p* = 0.03)) but did not vary at greater concentrations (log-rank, *p* = 0.32). (**B**) Inhibition of pirlimycin varied among pathogen groups (*p* < 0.009).

**Figure 2 antibiotics-13-00091-f002:**
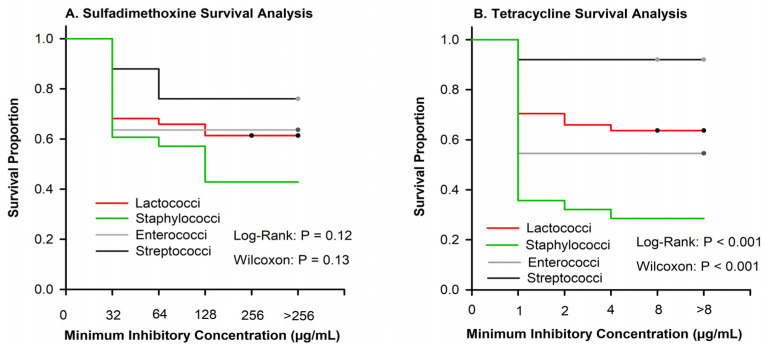
Kaplan–Meier plots showing the survival proportion of 108 isolates stratified by *Lactococci* (n = 44), *Enterococci* (n = 11), *Staphylococci* (n = 28), and *Streptococci* (n = 25). (**A**) Sulfadimethoxine curves showed no significant difference between organisms when evaluated using log-rank or Wilcox. (**B**) Tetracycline survival plot depicting significant differences between organisms using both log-rank and Wilcox *p* < 0.001.

**Figure 3 antibiotics-13-00091-f003:**
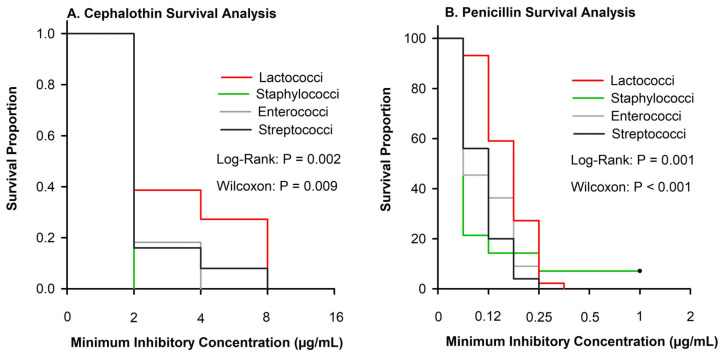
Kaplan–Meier plots showing the survival proportions of 108 isolates stratified by *Lactococci* (n = 44), *Enterococci* (n = 11), *Staphylococci* (n = 28), and *Streptococci* (n = 25). (**A**) Cephalothin curves showed significant difference between organisms when evaluated using log-rank or Wilcox *p* ≤ 0.009. (**B**) Penicillin survival plot depicting significant differences between organisms between log-rank or Wilcox *p* ≤ 0.001.

**Table 1 antibiotics-13-00091-t001:** Gram-positive organisms isolated from the quarters of cows with non-severe clinical mastitis cases occurring on 3 Michigan dairy farms from June 2019 to March 2020 and used for susceptibility testing (>10 isolates per pathogen group).

Pathogen Group	Number	Percent
Non-Aureus Staphylococci	28	25.9
*Staphylococcus chromogenes*	18	16.7
*Staphylococcus simulans*	5	4.6
*Staphylococcus haemolyticus*	2	1.8
*Staphylococcus species*	2	1.8
*Staphylococcus epidermidis*	1	0.9
*Streptococci*	25	23.1
*Streptococcus dysgalactiae*	16	14.8
*Streptococcus uberis*	5	4.6
*Streptococcus species*	4	3.7
*Lactococci*	44	40.7
*Lactococcus lactis*	26	24.1
*Lactococcus garvieae*	18	16.7
*Enterococci*	11	10.2
*Enterococcus saccarolyticus*	9	8.3
*Enterococcus faecalis*	1	0.9
*Enterococcus aquimarinus*	1	0.9
Total	108	100%

**Table 2 antibiotics-13-00091-t002:** Distribution of minimum inhibitory concentrations (MICs) from clinical mastitis collected from Michigan dairy farms between June 2019 and March 2020.

Drug	Etiology	B.P ^1^	Number	Sus. ^2^	Percent of Isolates at Each Indicated MIC (μg/mL)								N.I. ^3^
				%	0.12	0.25	0.5	1	2	4	8	16	%
Ampicillin †	NAS	≤0.12 ^4^	28	82.1	**82.1**	0.0	3.6	7.1	3.6	0.0	0	-	0.0
	Strepto.	≤0.25 ^4^	25	80.0	**64.0**	16.0	16.0	0.0	4.0	0.0	0.0	-	0.0
	Lacto.	≤0.25	44	63.7	45.5	**18.2**	34.1	2.3	0.0	0.0	0.0	-	0.0
	Entero.	≤8 ^4^	11	100.0	**81.8**	9.1	9.1	0.0	0.0	0.0	0.0	-	0.0
Ceftiofur *†	NAS	≤2	28	92.9	-	-	25.0	**39.3**	26.6	7.1	0.0	-	0.0
	Strepto.	≤2	25	100.0	-	-	**76.0**	24.0	0.0	0.0	0.0	-	0.0
	Lacto.	≤2	44	100.0	-	-	** 93.2 **	4.6	2.3	0.0	0.0	-	0.0
	Entero.	≤2 ^5^	11	100.0	-	-	**63.6**	27.3	9.1	0.0	0.0	-	0.0
Cephalothin *†	NAS	≤8 ^4^	28	100.0	-	-	-	-	** 100 **	0.0	0.0	0.0	0.0
	Strepto.	≤8 ^4^	25	100.0	-	-	-	-	**84.0**	8.0	8.0	0.0	0.0
	Lacto.	≤8	44	100.0	-	-	-	-	**61.4**	11.4	27.3	0.0	0.0
	Entero.	≤8 ^5^	11	100.0	-	-	-	-	**81.8**	18.2	0.0	0.0	0.0
Erythromycin *†	NAS	≤0.5 ^4^	28	89.2	-	32.1	**57.1**	3.6	0.0	0.0	-	-	7.1
	Strepto.	≤0.25 ^4^	25	80.0	-	**80.0**	0	4.0	4.0	0.0	-	-	12.0
	Lacto.	≤0.25	44	97.7	-	** 97.7 **	2.3	0.0	0.0	0.0	-	-	0.0
	Entero.	≤0.5 ^4^	11	100.0	-	** 100 **	0.0	0.0	0.0	0.0	-	-	
Oxacillin	NAS	≤2 ^4^	28	100.0	-	-	-	-	** 100 **	0.0	-	-	0.0
	Strepto.	≤2 ^4^	25	100.0	-	-	-	-	** 100 **	0.0	-	-	0.0
	Lacto.	≤2	44	100.0	-	-	-	-	** 100 **	0.0	-	-	0.0
	Entero.	≤2 ^5^	11		-	-	-	-	** 100 **	0.0	-	-	0.0
Penicillin *†	NAS	≤0.12 ^4^	28	78.6	**78.6**	7.1	0.0	7.1	0.00	-	-	-	7.1
	Strepto.	≤0.12 ^4^	25	44.0	44.0	**36.0**	16.0	4.0	0.00	-	-	-	0.0
	Lacto.	≤0.12	44	6.8	6.8	34.1	**31.8**	25.0	2.3	-	-	-	0.0
	Entero.	≤8 ^4^	11	100.0	**54.6**	9.1	27.3	9.1	0.00	-	-	-	0.0
Penicillin novobiocin ^7^	NAS	≤1/2 ^6^	28	96.4	-	-	-	-	** 96.4 **	0.0	0.0	0.0	3.6
	Strepto.	≤1/2	25	100.0	-	-	-	-	** 100 **	0.0	0.0	0.0	0.0
	Lacto.	≤1/2	44	100.0	-	-	-	-	** 100 **	0.0	0.0	0.0	0.0
	Entero.	≤1/2 ^5^	11	100.0	-	-	-	-	** 100 **	0.0	0.0	0.0	0.0
Pirlimycin *†	NAS	≤2 ^6^	28	92.9	-	-	**85.7**	3.6	3.6	3.6	-	-	3.6
	Strepto.	≤2	25	84.0	-	-	**76.0**	4.0	4.0	0.0	-	-	16.0
	Lacto.	≤2	44	59.1	-	-	**52.3**	0.0	6.8	2.3	-	-	38.6
	Entero.	≤2 ^5^	11	90.6	-	-	**63.3**	18.2	9.1	9.1	-	-	0.0
Tetracycline *†	NAS	≤4 ^4^	28	71.5	-	-	-	**64.3**	3.6	3.6	3.6	-	25.0
	Strepto.	≤2 ^4^	25	8.0	-	-	-	8.0	0.0	16.0	0.0	-	76.0
	Lacto.	≤2	44	34.2	-	-	-	29.6	4.6	2.3	4.6	-	59.1
	Entero	≤4 ^4^	11	45.0	-	-	-	45.0	0.0	0.0	0.0	-	55.0
Sulfadimethoxine ^8^	NAS	≤128 ^9^	28	57.2	-	-	-	-	39.3	3.6	**14.3**	0.0	39.3
	Strep	≤128 ^9^	25	24.0	-	-	-	-	12.0	12.0	0.0	0.0	76.0
	Lacto.	≤128 ^9^	44	38.6	-	-	-	-	31.8	2.3	4.6	2.3	59.1
	Entero	≤128 ^9^	11	36.4	-	-	-	-	36.6	0.0	0.0	0.0	63.6

^1^ B.P indicates the breakpoint at which an isolate is considered susceptible according to CLSI Vet01S-Ed5. As there is no B.P for *Lactococcus* spp. in CLSI Vet01S-Ed5, the breakpoint for *Streptococci* was used. ^2^ Percentage of susceptible isolates based on the breakpoints in CLSI Vet01S-Ed5 [12,13]. ^3^ Isolates that were not inhibited at the greatest concentration of the antimicrobial tested. ^4^ No breakpoint for bovine mastitis organism exists in CLSI; interpretative criteria are from humans [12,13]. ^5^ No breakpoint for mastitis pathogens for *Enterococci*; the B.P is from *Streptococci* was used [12,13]. ^6^ No B.P available for mastitis pathogens for NAS; the B.P from *Staph aureus* for mastitis was used [12,13]. ^7^ Tested concentrations of Penicillin Novobiocin were 1/2, 2/4, 4/8, and 8/16, represented in the table by the concentration in µg/mL of Novobiocin. ^8^ Tested concentrations of Sulfadimethoxine were 32, 64, 128, and 256, which are represented in the table as 2, 4, 8, and 16 µg/mL, respectively. ^9^ The current CLSI guidelines do not have recommendations for sulfadimethoxine; for any human or bovine mastitis pathogen to be consistent with the previous literature, susceptibility was defined as ≤128 µg/mL [12,13,14]. **Bold** indicates MIC_50_, while underlined indicates MIC_90_. - Indicates values not tested for the indicated antimicrobial. * Log-rank test for equality of strata (pathogen) at higher minimum inhibitory concentration was significant *p* < 0.002. † Wilcoxon test for equality of strata (pathogen) at lower minimum inhibitory concentrations was significant *p* < 0.009.

## Data Availability

The data presented in this study are available on request from the corresponding author. The data are not publicly available due to data originating from privately held animals.

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
