# Peer review of "Comparison of Minimum Inhibitory Concentrations of Selected Antimicrobials for Non-Aureus Staphylococci, Enterococci, Lactococci, and Streptococci Isolated from Milk Samples of Cows with Clinical Mastitis"

_antibiotics, 2024, doi:10.3390/antibiotics13010091_

Round 1
Reviewer 1 Report
Comments and Suggestions for Authors
In recent years, mastitis due to environmental pathogens has increased. Non-aureus staphylococci and mammaliicocci (NASM) are one of the most common causes of subclinical mastitis in dairy animals and the extent of damage by intramammary infections (IMI) caused by NASM is still under debate. The different effects of NASM on the mammary gland may be associated with differences between bacterial species. NASM are normal and abundant colonizers of humans and animals and become pathogenic only in certain situations.MALDI-TOF simplify the routine differentiation of esculinhydrolyzing streptococci and streptococcal-like organisms.
This is a well and concisely written study and therefore I have no major comments or complaints from my professional perspective. I'm not sure if a work conceived in this way meets the requirements to be characterized as an Article or if it could rather be classified as a Case report or Communication.
MALDITOFF, IMM - at the first appearance of the abbreviation, the full name should be given.
Farm in MI and Minesota... MI?
Describe in detail the method of sampling and diagnostic of clinical mastitis.
Author Response
Dear reviewer,
Thank you for your thoughtful comments. Our responses to your comments are below.
REV: This is a well and concisely written study and therefore I have no major comments or complaints from my professional perspective. I'm not sure if a work conceived in this way meets the requirements to be characterized as an Article or if it could rather be classified as a Case report or Communication.
AU: Thank you for your kind words. We wondered about the classification as well but were told by the editor to submit as an article. We will leave as such unless the editors wish otherwise.
REV: MALDITOFF, IMM - at the first appearance of the abbreviation, the full name should be given.
AU: We made this change at line 168. Thank you. We did not change in the abstract but will do so if the editors suggest.
REV: Farm in MI and Minesota... MI?
AU: Thank you for catching this. We corrected the error and spelled out Michigan.
REV: Describe in detail the method of sampling and diagnostic of clinical mastitis.
AU: we have added more detail about sampling and detection of CM.
Reviewer 2 Report
Comments and Suggestions for Authors
In the manuscript are some minor comments.
Overall is interesting but is a very common theme, and therefore the introduction and discussion should be improved. Materials and methods also should be improved.
check references

Author Response
Thank you for the time that you spent reviewing our paper and helping us to improve it. We appreciate your time and have made the corrections that you suggest. We hope that the paper is now suitable for publication.
Line 1-3, and lines 20-21: italics added as suggested throughout the paper. These changes were frequent so were not highlighted.
As the reviewer suggested, some changes were made to the introduction and discussion and edits have been made to the material and methods. This paper is submitted to a special issue about this topic, so we did not extensively add justification for performing the study.
All the references have been reviewed for consistency in formatting and to be sure that they are appropriate. Some changes have been made and additional references added.
We hope that these changes satisfy the concerns of this reviewer and are appreciative of the time that you spent reading our paper.
Reviewer 3 Report
Comments and Suggestions for Authors
Dear Editor
The manuscript "Comparison of minimum inhibitory concentrations of selected antimicrobials for non-aureus Staphylococci, Enterococci, Lactococci, and Streptococci isolated from milk samples of cows with clinical mastitis" is well conducted and presented study that could assist the researchers regarding MIC of selected antimicrobial against NAS and other Gram positive cocci. However some of the queries or suggestion need to be addressed prior to the final decision:
Line 23-24: "Minimum inhibitory concentrations (MIC) were determined using the mastitis panel of a commercial broth microdilution test". The authors are suggested to mention any conflict in the methods sections regarding this.
Line 32: This is confusing statement "-----, indicating that pathogen differences in MIC are not likely clinically relevant".
Line 34-35: This is not necessary to indicate specifically "in the United States" as the clinical situation and response is similar to great extent across the globe.
Line 42-44: The Authors are suggested to indicate 2-3 examples of CIA along with, if the current US Food and Drug Administration Reference is available.
Line 52-55: The Authors are suggested to indicate one global reference along with ref 9.
Line 55-58: the authors are suggested to include one more objective as "the identification of NAS and other cocci". This identification is a significant part of the study.
Results: The description of results is significantly overlapping as text and Table 1 and Table 2. The Authors are suggested to reduce the text as they have very well explanation in tables.
Line 135: "with previous literature susceptibility was defined as <128 μg/mL". The authors are suggested to cite the reference here.
Line 171-182: are these lines belonged to Author's knowledge or otherwise need references?
Line 339-341: If the farmers are performing the test at farms, the authors could mention here, the line of action of farmers.
Thanks and Regards
Comments on the Quality of English Language
The quality of the language is very good with few minor errors.
Author Response
Dear Reviewer,
Thank you for your help in improving this manuscript. We have made the suggested changes and hope that the paper is now acceptable.
Line 23-24: "Minimum inhibitory concentrations (MIC) were determined using the mastitis panel of a commercial broth microdilution test". The authors are suggested to mention any conflict in the methods sections regarding this.
AU: We are not very sure what conflicts the reviewer is referring to. We changed the wording on line 24 to be more clear that we used a commercially available test. The test that we used is the gold standard for MIC testing and includes all approved antibiotics available in the US. It is the only test marketed for this purpose in the US.
REV: Line 32: This is confusing statement "-----, indicating that pathogen differences in MIC are not likely clinically relevant".
AU: We added some clarifying text to line 32 (."..as these are the two most commonly administered mastitis treatments in teh United States."
REV: Line 34-35: This is not necessary to indicate specifically "in the United States" as the clinical situation and response is similar to great extent across the globe.
AU: we deleted United States as suggested.
REV: Line 42-44: The Authors are suggested to indicate 2-3 examples of CIA along with, if the current US Food and Drug Administration Reference is available.
AU: Thank you for this suggestion. The FDA reference was dated so we replaced it with the World Health Organization reference which is more current. We also specifically referred to 3rd generation cephalosporins (which are commonly used to treat mastitis in the US). (l43)
REV: Line 52-55: The Authors are suggested to indicate one global reference along with ref 9.
AU: We have added Plumed-Ferrer et al., Vet Micro 2013 L55
REV: Line 55-58: the authors are suggested to include one more objective as "the identification of NAS and other cocci". This identification is a significant part of the study.
AU: done as suggested. Thank you. L60
REV: Results: The description of results is significantly overlapping as text and Table 1 and Table 2. The Authors are suggested to reduce the text as they have very well explanation in tables.
AU: We made some changes to reduce redundancy. Thank you.
REV: Line 135: "with previous literature susceptibility was defined as <128 μg/mL". The authors are suggested to cite the reference here.
AU: done as suggested. (L140)
REV: Line 171-182: are these lines belonged to Author's knowledge or otherwise need references?
AU: I don't really know how to provide a reference for these statements. Basically, there are 5 IMM products (only IMM drugs are approved for treatment of mastitis in the US) and to make this statement, we simply read all the product labels by googling each product. We didn't make any changes here as there is no concise place to find this information. Sorry.
REV: Line 339-341: If the farmers are performing the test at farms, the authors could mention here, the line of action of farmers.
AU: clarifying text was added. Thank you. L356
Round 2
Reviewer 2 Report
Comments and Suggestions for Authors
All the corrections were performed